# Antimicrobial Resistance Profiles of *Staphylococcus* Isolated from Cows with Subclinical Mastitis: Do Strains from the Environment and from Humans Contribute to the Dissemination of Resistance among Bacteria on Dairy Farms in Colombia?

**DOI:** 10.3390/antibiotics12111574

**Published:** 2023-10-28

**Authors:** Martha Fabiola Rodríguez, Arlen Patricia Gomez, Andres Ceballos-Garzon

**Affiliations:** 1School of Health Sciences, Universidad de La Salle, Bogotá 110421, Colombia; 2Facultad de Medicina Veterinaria y de Zootecnia, Universidad Nacional de Colombia, Sede Bogotá, Bogotá 111321, Colombia; apgomezr@unal.edu.co; 3Proteomics and Human Mycosis Unit, Infectious Diseases Group, Microbiology Department, School of Science, Pontificia Universidad Javeriana, Bogotá 110231, Colombia; c-ceballos@javeriana.edu.co; 4BIOASTER, Institut de Recherche Technologique, 40 Avenue Tony Garnier, 69007 Lyon, France

**Keywords:** antimicrobial resistance, dairy farm, *Staphylococcus*, subclinical mastitis

## Abstract

*Staphylococcus* is a very prevalent etiologic agent of bovine mastitis, and antibiotic resistance contributes to the successful colonization and dissemination of these bacteria in different environments and hosts on dairy farms. This study aimed to identify the antimicrobial resistance (AMR) genotypes and phenotypes of *Staphylococcus* spp. isolates from different sources on dairy farms and their relationship with the use of antibiotics. An antimicrobial susceptibility test was performed on 349 *Staphylococcus* strains (*S. aureus*, *n* = 152; non-aureus staphylococci (NAS), *n* = 197) isolated from quarter milk samples (QMSs) from cows with subclinical mastitis (176), the teats of cows (116), the milking parlor environment (32), and the nasal cavities of milk workers (25). Resistance and multidrug resistance percentages found for *S. aureus* and NAS were (*S. aureus* = 63.2%, NAS = 55.8%) and (*S. aureus* = 4.6%, NAS = 11.7%), respectively. *S. aureus* and NAS isolates showed resistance mainly to penicillin (10 IU) (54.1% and 32.4%) and ampicillin (10 mg) (50.3% and 27.0%) drugs. The prevalence of AMR *Staphylococcus* was higher in environmental samples (81.3%) compared to other sources (52.6–76.0%). In *S. aureus* isolates, the identification of the *blaZ* (83.9%), *aacAaphD* (48.6%), *ermC* (23.5%), *tetK* (12.9%), and *mecA* (12.1%) genes did not entirely agree with the AMR phenotype. We conclude that the use of β-lactam antibiotics influences the expression of AMR in *Staphylococcus* circulating on dairy farms and that *S. aureus* isolates from the environment and humans may be reservoirs of AMR for other bacteria on dairy farms.

## 1. Introduction

Staphylococci comprise many species that colonize the skin and mucosa of various mammalian hosts [1,2]. A complex network of regulatory signals facilitates their adaptation and establishment in available niches throughout their life cycle [3]. *S. aureus* is the most pathogenic species of the genus and is easily transmitted between animals and humans, constituting a serious threat to animal, human, and public health [4]. In cattle, *S. aureus* is a “major pathogen” very prevalent in bovine mastitis in the world [5,6,7,8], and non-aureus staphylococci (NAS) species, such as *S. simulans*, *S. chromogens*, *S. hyicus*, and *S. epidermis*, are “minor pathogens” considered emerging pathogens, frequently isolated from quarter milk samples (QMSs) from cows with subclinical mastitis in many countries [9,10]. Genetic studies have shown that some clones are more persistent within and among dairy herds [4,11], likely because they carry virulence and antibiotic resistance genes that contribute to the successful colonization and dissemination of bacteria in different environments and hosts on dairy farms [12,13].

The main strategy to treat mastitis is through the use of antibiotics (penicillin, ampicillin, tetracycline, gentamicin, etc.) [14]. β-Lactams continue to be the most widely employed, especially penicillin and cephalosporins, even though bacteria rapidly develop resistance to them [15,16]. One of the most common resistance mechanisms is the production of β-lactamases, encoded by the *blaZ* gene, whose transmission is influenced by the selective pressure of antibiotic use [17]. The prevalence of *blaZ* differs according to the geographical region, with large differences found in places where *S. aureus* isolates have been obtained from cows with mastitis [18]. Some studies claim that the gene is not transferred across different *Staphylococcus* species of animal origin, even when they are present in the same herd [19]. Therefore, it is still controversial whether the *blaZ* gene is transmitted vertically (clonally) or horizontally in this genus of bacteria [20].

Additionally, the *Staphylococcus* genus can also carry the *mecA* gene that encodes the penicillin-binding protein 2a variant (PBP2a), which has a low affinity for β-lactams, conferring resistance to these antibiotics, including penicillinase-resistant penicillins [21]. The detection of the *mecA* gene in methicillin-resistant *S. aureus* and NAS strains (MRSA and SMR, respectively) isolated from cows with mastitis has been associated with human contact, as well as poor hygiene in milking equipment [22]. The MRSA spread in the community (CA-MRSA), hospitals (HA-MRSA), and cattle (LA-MRSA) is also likely mediated by virulence factors that allow adaptation and colonization in different hosts, increasing the zoonotic risk. Some lineages, such as C22, CC8, and CC398, can be transmitted between humans and animals [23]. In Colombia, the *blaZ* and *mecA* genes have been identified in 59 and 26% of *Staphylococcus* strains obtained from bovine milk, respectively [24]. 

Regardless of the presence or absence of the *mecA* gene, the susceptibility of the *Staphylococcus* genus to β-lactam antibiotics varies widely among geographic regions, depending on the circulating strains. In Europe and the USA, the majority of *Staphylococcus* spp. isolated from cows with mastitis are susceptible to these antibiotics; less than 50% are resistant to penicillin and ampicillin, with Switzerland and Sweden being the countries where the lowest percentage of resistance to these antibiotics has been reported [25,26]. Conversely, in China, more than 80% of *Staphylococcus* spp. isolated from cases of bovine mastitis are resistant to penicillin and ampicillin, as well as aminoglycosides, erythromycin, and tetracycline [27,28]. In South Africa, a high prevalence of resistance to penicillin (50.3%) and tylosin (67.1%) has also been reported [29]. In Algeria, North Africa, between 71.4% and 100.0% of *Staphylococcus* spp. isolates are resistant to penicillin, 54.3% are resistant to erythromycin, and 40.0% are resistant to tetracycline [30]. In South America, the highest resistance to penicillin and ampicillin in *Staphylococcus* spp. isolates from cows with mastitis has been reported in Mexico (over 85.0%), Paraguay (over 70.0%), and Argentina (50.0%) [9,31,32,33]. In Colombia, most strains of *Staphylococcus* isolated from cows with mastitis are sensitive to antibiotics commonly used to treat infections. Resistance to penicillin (20.0–28.7%), ampicillin (24.8%), tetracycline, and erythromycin (14.9%) has been reported [24,34].

The ease with which *Staphylococcus* spp. isolates from dairy farms acquire AMR has been attributed to selective pressure for resistant strains that are difficult to eradicate due to the lack of surveillance by mastitis prevention and control programs and to the empirical use of β-lactam antibiotics in humans and animals [35,36]. Other factors that also contribute to the appearance and spread of AMR in agricultural production systems include poor storage of drugs, lack of prescription of antibiotics by veterinarians, lack of programs to reduce the risk of the introduction and spread of infections on farms, environmental contamination with excreted antimicrobials or their metabolites, antimicrobial residues in edible tissues, and direct zoonotic transmission [37,38]. The study of AMR profiles of pathogenic, commensal, and opportunistic *Staphylococcus* species is an indirect indicator of these factors and of the plasticity of bacteria to adapt, compete, and survive under adverse conditions in a geographic niche. Therefore, the purpose of this research was to identify the AMR genotypes and phenotypes of *Staphylococcus* spp. isolates from different sources on dairy farms and their relationship with the use of antibiotics for treating mastitis.

## 2. Results

### 2.1. Data on Farms, Prevention, and Treatment of Bovine Mastitis

Of the thirteen farms that were enrolled, four (30.8%) were from the small tier, five (38.5%) were from the medium tier, and four (30.8%) were from the large tier. Most farms (*n* = 11) used penicillins (penicillin, cloxacillin, ampicillin, and amoxycillin) and cephalosporins for cow dry-off treatment; only two farms (15.4%) did not use antibiotics for dry-offs. The verification of the percentage of compliance with good farming practices and their relationship (*χ*^2^) with resistance to one or more antibiotics revealed critical points in the pharmacological treatment of mastitis (prescription of antibiotics by veterinarians, withdrawal periods of antibiotics, storage and inventory of medicines): 68.0% (*χ*^2^: 37.6; *p* = 0.000); animal health (mastitis prevention and control program, cultures for bacterial identification and antibiogram): 44.2% (*χ*^2^: 23.9; *p* = 0.000); facilities and hygiene of the milking parlor: 73.0% (*χ*^2^: 33.4; *p* = 0.000); staff training, staffing, and cleanliness: 28.2% (*χ*^2^: 20.2; *p* = 0.000); good practices during the milking routine: 77.0% (*χ*^2^: 18.9; *p* = 0.000); animal hygiene (udders, tails, and flanks): 44.2% (*χ*^2^: 12.5; *p* = 0.001); and the cleanliness of the milking equipment: 64.1% (*χ*^2^: 7.2; *p* = 0.03) (Table 1).

An association was found between the percentage of use of β-lactams in the treatment and prevention of bovine mastitis and the high rate of AMR of these isolates to penicillins (*p* < 0.05). On the farms on which penicillins or cephalosporins were not used, the lowest percentages of AMR to penicillin (22.9%) and ampicillin (29.4%) were found, and in those that used cephalosporins (without penicillins), the highest percentages of AMR to penicillin (61.1%) and ampicillin (54.2%) were obtained. However, the analysis of AMR per farm did not show a relationship between the therapeutic use of a specific group of antibiotics and the resistance found in vitro to the same antibiotic.

### 2.2. Microbiological Identification

A total of 349 isolates (*S. aureus*, *n* = 152 (43%); NAS, *n* = 197 (57%)) were obtained from milk samples, *n* = 176 (50.4%) (*S. aureus*, 117; NAS, 59), from cows with subclinical mastitis; the skin of the cows’ teats, *n* = 116 (33.2%) (*S. aureus*, 18; and NAS, 98); the environment, *n* = 32 (9.2%) (*S. aureus*, 7; NAS, 25); and from workers’ nasal mucosa, *n* = 25 (*S. aureus*, 10; NAS 15). The most frequently isolated species belonging to the NAS group were *S. chromogenes* (41/197), *S. haemolyticus* (37/197), and *S. epidermidis* (25/197). Other NAS (94/197) identified were *S. warneri*, *S hominis*, *S. hyicus*, *S. lentuns*, *S. equorum*, *S. auricularis*, *S. arlettae*, *S. lutetiensis*, *S. lugdunensis*, *S. capitas*, *S. sciuri*, *S. xylosus*, and *S. vitulinus*. Although *S. aureus* was found in all samples tested, milk samples were the main source (i.e., QMSs: 117; teats: 116; environmental: 32; milkers: 25).

### 2.3. Antimicrobial Resistance Phenotype

To determine the antimicrobial susceptibility profile of all staphylococci isolates, the Kirby–Bauer disk diffusion method was performed. The vast majority of *S. aureus* isolates showed susceptibility to all antibiotics tested, except penicillin and ampicillin, with resistance of 54% and 50%, respectively. Resistance to amoxycillin reached 13%; for all other molecules, resistance rates were below 10%. Similarly, all isolates from the NAS group showed low rates of resistance to all antibiotics evaluated. Resistance to penicillin and ampicillin reached 32% and 27%, respectively. The antibiotics for which the lowest percentages of resistance were observed for both *S. aureus* and NAS were cefoperazone and gentamicin. Among 152 *S. aureus* isolates, 96 (63.2%) were resistant to at least one antibiotic, of which seven (4.6%) were multidrug-resistant (resistant to three or more antibiotic classes) (Table 2).

*S. epidermidis* was the species with the highest percentage of AMR, mainly against penicillin (69.6%), ampicillin (52.2%), and erythromycin (50.0%), with significant differences (*p* < 0.05) from the resistance rates found in the other NAS species (Figure 1).

According to the source of isolation, the highest percentage of AMR found in *S. aureus* was in the nasal mucosa of the workers. In these isolates, the greatest resistance was to penicillin (77.8%), ampicillin (66.7%), tetracyclines (66.3%), amoxycillin (50.0%), and cefoxitin (40.0%), followed by isolates of environmental origin, which were mainly resistant to ampicillin (100.0%), penicillin (66.7%), and amoxycillin (40.0%). These isolates showed a significant increase (*p* < 0.05) in resistance to β-lactam antibiotics. The NAS isolates with AMR identified in the different sources were mainly resistant to β-lactam antibiotics, tetracyclines, and trimethoprim/sulfamethoxazole; there were no significant differences (*p* > 0.05) in AMR among groups (Table 3).

Forty-one percent (142/349) of the total *Staphylococcus* spp. were susceptible, 59.0% (206/349) presented resistance to one or two groups of antibiotics, and 8.6% (30/349) were multiresistant, of which eleven were obtained from cows’ milk with subclinical mastitis, nine were from the skin of the teats, six were from the nasal mucosa of the workers, and four were from the environment. *S. epidermidis* was the species with the highest number of multiresistant isolates (*n* = 11). No statistically significant association (*p* > 0.05) was established between the number of multiresistant *Staphylococcus* and the source of the isolates (Table 4).

### 2.4. S. aureus Antimicrobial Resistance Genotype

Eighty-three-point-nine percent of *S. aureus* isolates had the *blaZ* gene. Of these, 52.5% expressed resistance to any of the penicillins evaluated by the Kirby–Bauer technique. The *mecA* gene was detected in 17 isolates, of which only 3 had the MRSA phenotype, as determined with the cefoxitin disk. The other gene that was present with a high frequency (48.6%) in *S. aureus* was *aacA-aphD*. However, the expression of resistance to gentamicin was very low (1.7%), and only one isolate presented a coincidence between the genotype and phenotype. Overall, there was no agreement between the phenotype and genotype, with a Cohen’s Kappa coefficient <0.16 (Table 5).

## 3. Discussion

On the dairy farms of the Bogotá savanna, *S. aureus* was the most isolated species from cows with subclinical mastitis and other sources. The ability of *S. aureus* to survive in different animal and human hosts and in food, soil, and air has been explained by a complex network of regulatory genes that control the expression of virulence factors and AMR [39,40,41]. In this research, more than 80% of *S. aureus* isolated from different sources carried the *blaZ* gene and, to a lesser extent, the *aacA-aphD*, *emrC*, *tetK*, and *mecA* genes. In all cases, the expression of these genes was significantly lower, and a good correspondence was only found between the penicillin resistance genotype and phenotype, which was the most frequent in the *Staphylococcus* analyzed. The high global AMR of *Staphylococcus* to β-lactams has led to the selection of resistant *S. aureus* strains that are difficult to eradicate [28,39,42]. This has been attributed to unattended exposure to these antibiotics in bovine mastitis prevention and control programs [39,40,43]. The percentage of non-compliance (32.0%) with the prescription of antibiotics by veterinarians and the withdrawal times of antibiotics on the farms studied were significantly associated with the percentage of AMR in *Staphylococcus* isolates. Additionally, of the 13 dairy farms studied, 11 included some type of penicillin or cephalosporin for the treatment or prevention of bovine mastitis, which was significantly associated with resistance to penicillin and ampicillin expressed by *S. aureus.*

In contrast, more than 90.0% of the *S. aureus* isolates were sensitive to gentamicin, erythromycin, and tetracycline, despite the fact that some carried the corresponding AMR genes. Possibly, the infrequent use of these antibiotics has influenced the low expression of the respective AMR genes. In Colombia, *S. aureus* isolates obtained from bovine milk between 2010 and 2014 also showed resistance to penicillin and ampicillin, although in lower percentages (23.1% and 28.5%, respectively). Similar to our results, a significant number of the isolates that had resistance genes for β-lactams, erythromycin, tetracycline, and aminoglycosides did not show phenotypic resistance to the respective antibiotic [24]. 

The study of the location of the *blaZ* gene in the *Staphylococcus* genus has revealed that bovine isolates carry the gene mainly on the chromosome rather than on plasmids, as occurs with human isolates; apparently, the *blaZ* gene in the different species of *Staphylococcus* obtained from animals and humans has a common ancestor, independent of the location of the gene. However, the plasmids that carry it are different between *S. aureus* and NAS, so it is unlikely that the *blaZ* gene will be exchanged between these species, which is independent of the successful dissemination of AMR throughout the world [17]. The latter will depend on the clonal dissemination of a strain among bovines, regardless of its origin (human or bovine), which has been successfully adapted in a herd, explaining why, despite the genetic variability of the strains, there are few clones that are predominant.

The lack of correlation between the AMR genotype and phenotype has also been reported in other research [18,44]. AMR genes can be acquired by vertical or horizontal transfer between strains of different species isolated from various niches, giving bacteria an evolutionary advantage over the use of antimicrobials [21,45]. Thus, a population of bacteria can lose the resistance phenotype in the absence of exposure to the antibiotic, conserving the gene that confers said resistance [46], which would explain the lack of correspondence between the AMR genotype and phenotype in this and other studies. 

Another interesting finding in this research was that the highest AMR was found in isolates from the environment and humans, mainly against penicillins, cefoxitin, erythromycin, and tetracyclines, being significantly higher in *S. aureus*. These results are similar to those reported for NAS and MRSA of human and environmental origin on dairy farms in different countries [47,48]. Other studies have found that isolates of *S. aureus* and NAS from environmental sources (milking parlors, canteens, and cooling tanks, among others) and from humans share the same AMR and virulence profiles with strains isolated from the milk of cows with mastitis [49,50]. Although the identification of genetic variants was not carried out in this study, these results suggest a possible transmission of multiresistant *Staphylococcus* that is difficult to eradicate from humans to animals. Thus, cows with chronic subclinical mastitis that acquire persistent *S. aureus* strains become the most important reservoirs in the herd [11]. Similarly, isolates from the environment with high AMR could serve as reservoirs of AMR and virulence genes [47,51], as well as NAS species that are part of the udder skin microbiota [52,53].

NAS species have been reported as important reservoirs of virulence genes and AMR [19,54]. In this regard, it is important to highlight that the AMR of the *S. epidermidis* isolates was significantly higher than in the other species. Several reports indicate that this species has acquired virulence and resistance genes over time, which have increased its pathogenic potential in different hosts [55,56]. Furthermore, *S. epidermidis* was the species with the highest number of multiresistant isolates from the four sources analyzed. Similar results have been found in workers and animals on dairy, meat, and poultry farms in Belgium [56], suggesting that *S. epidermidis* may act as a reservoir of AMR for other bacteria in these ecosystems. NAS multiresistance profiles were identified at all sampling sites and were resistant to more antibiotic groups than *S. aureus* isolates. The multiresistance of these bacteria mainly involved the groups of β-lactams, macrolides, and tetracyclines. This AMR profile has been reported in *Staphylococcus* isolated from the environment of pig farms in Germany [54], from workers, the environment, and animals of dairy farms in Italy [47], and milk from cows with subclinical mastitis in China [27], among others. The combined use of antibiotics, the acquisition and transport of AMR genes in the microbiota, and antibiotic residues in the farm environment also contribute to the selection of multiresistant strains [57]. Therefore, the geographic variability in resistance is directly related to the impact of control strategies and the treatment of infectious diseases, as well as the management and monitoring of these drugs. In the present study, the high percentage of farms (55.8%) without mastitis prevention and control programs and without cultures for the identification of the microorganism or antibiogram was significantly associated with AMR of *Staphylococcus* isolated.

The *mecA* gene was detected in 17 isolates, of which only 3 had an MRSA phenotype determined with the cefoxitin disk. None presented a multiresistance profile, and most only had resistance to penicillin, contrary to the AMR profiles of MRSA found in other countries [47,51,58]. MRSA was mainly isolated from milk samples from cows with subclinical mastitis, although the difference in the number of samples from the different isolates should be taken into account. In bovines, the ST8, ST130, and ST398 lineages of MRSA have been identified. The ST398 lineage has been identified as an emerging clone in bovines; its origin appears to be human, and it is easily transmitted between various species of animals and the workers in charge of their care [59]. The AMR profile of the ST398 lineage varies widely. In general, they are resistant to β-lactams, tetracyclines, and erythromycin. Like CA-MRSA, LA-MRSA isolates are susceptible to most antibiotics [60], as evidenced in this research.

## 4. Materials and Methods

### 4.1. Study Design and Bacterial Isolates

In 2018 and 2019, a field study was carried out on thirteen dairy farms located in the Bogotá savanna, with an altitude of 2600 m above sea level and an average temperature of 9 °C (range between −5 °C and 26 °C). Farms that had a mechanical milking system, registration of production, veterinary care, and good milking practices were selected and classified into the following tiers: small (10–35 bovines), medium (36–100 bovines), and large (more than 100 bovines). From a total of 1784 samples (330 cows), 349 isolates of staphylococci (*S. aureus*: 152; NAS: 179) were obtained from the skin of the apex of the teats (*S. aureus*: 18; NAS: 98); from quarter milk samples (QMSs) from cows with subclinical mastitis (*S. aureus:* 117; NAS: 59), classified according to the California Mastitis Test (CMT) [61]; from the environment (*S. aureus*: 7; NAS: 25); and from the workers’ nasal mucosa (*S. aureus*: 10; NAS: 15). Samples were streaked on blood agar and MacConkey agar plates (Difco Laboratories) and incubated at 37 °C for 24 to 48 h. Isolates were identified by MALDI-TOF MS (Bruker Daltonik, Bremen, Germany) according to the Bruker Daltonics protocol [62]. Briefly, one colony from trypticase soy agar media (TSA) (previously incubated at 37 °C for 18 to 24 h) was spotted onto a 96-spot steel plate (Bruker Daltonik, Bremen, Germany) and allowed to dry at room temperature before the addition of l μL of formic acid and HCCA matrix (provided by the supplier). Each colony was tested in duplicate. Only the spot returning the highest probability score of identification was considered. Protein mass spectra were analyzed using Flex Control^®^ software and MALDI Biotyper version 3.1 7311 reference spectra (main spectra) (Bruker Daltonics, Bremen, Germany) [62]. MALDI-TOF MS results were analyzed according to the manufacturer’s technical specifications as follows: correct identification of genus and species (≥2.0), correct identification of genus (1.7–2.0), and no reliable identification (<1.7) [62]. The owner or administrator filled out a questionnaire form and indicated the dates of prevention and pharmacological treatment of mastitis, as well as compliance with milking routine, hygiene and biosecurity rules, handling of abnormal milk, staff training, and storage and inventory of medicines.

### 4.2. Antimicrobial Susceptibility Test

The antimicrobial susceptibility test was performed using the Kirby–Bauer disk diffusion method, according to the Clinical and Laboratory Standards Institute (CLSI) guidelines [63]. Briefly, bacteria previously identified were spiked onto TSA and incubated at 37 °C for 24 h. The isolated colonies were re-suspended in a sterile saline solution at a concentration of 1.5 × 10^8^ bacteria/mL (corresponding to 0.5 McFarland standard measured with DensiCHECK plus BioMerieux, Marcy l’Etoile, France). Muller–Hinton agar (Difco Laboratories) plates were inoculated by evenly swabbing with a sterile cotton swab, and the following antibiotic disks were placed: cefoxitin (30 mg), oxacillin (1 mg), penicillin G (10 IU), ampicillin (10 mg), amoxycillin (25 mg), cefoperazone (75 mg), cephalothin (30 mg), ceftiofur (30 mg), erythromycin (15 mg), tetracycline (30 mg), ciprofloxacin (5 mg), gentamicin (10 mg), and sulfamethoxazole/trimethoprim (25 mg) (Oxoid Ltd., Basingstoke, UK). The plates were incubated at 37 °C overnight. Inhibition diameters were read with reflected light and reported in mm. The sizes of the zones of inhibition were interpreted as R (resistant), I (intermediate), and S (susceptible), taking into account the breakpoints reported by “Performance standards for antimicrobial disk and dilution susceptibility tests for bacteria isolated from animals” [64]. Two reference strains of *S. aureus*-positive (ATCC^®^ 43300) and *S. aureus*-negative (ATCC^®^ 25923) for the *mecA-1* gene were used as controls.

### 4.3. Identification of AMR Genes

*S. aureus* isolates underwent genomic DNA extraction and purification using the commercial PureLink™ Genomic DNA Mini Kit # K1820-01 (Thermo Fisher Scientific, Massachusetts, USA), following the manufacturer’s guidelines. DNA concentration and quality were measured with Nanodrop 2000 (Thermo Fisher Scientific, Waltham, MA, USA) and stored at −20 °C until use. Resistance genes to macrolides (*ermB*), tetracyclines (*tetK*), β-lactams (*blaZ*), methicillin (*mecA*), and aminoglycosides (*aacA-aphD*) were detected using the reported primers and PCR conditions (Table 6). The PCR was carried out in a final reaction volume of 25 μL. Each reaction mixture contained 12.5 μL of GoTaq^®^ Green Master Mix (2×) (Promega, Madison, WI, USA), 0.5 µL of each primer (0.1 µM), and 2 µL of bacterial DNA (10–30 ng). PCR products were visualized on a 1.5% agarose gel stained with HydraGreen^®^ (ACTGene, Piscataway, NJ, USA).

### 4.4. Statistical Analysis

The data obtained were tabulated and analyzed with descriptive statistics. For the survey, the percentage of compliance, milking conditions, and the prevention and pharmacological treatment of mastitis were determined by giving each question a value of 1 for compliance and 0 for non-compliance. The association or relationship of these variables with resistance to two or more antibiotics was determined with Pearson’s chi-square test of independence, and the agreement between phenotype and genotype was calculated with Cohen’s Kappa coefficient. All statistical analyses were performed considering a significance level of *p* < 0.05 using the R language. 

## 5. Conclusions

These results show the impact of the use of β-lactam antibiotics for the control and treatment of bovine mastitis on penicillin resistance in *Staphylococcus* strains circulating on dairy farms. Additionally, the high AMR of the isolates from the equipment and the milking parlor warns about the importance of controlling environmental contamination with antimicrobials and reiterates that the geographical variability in resistance is directly related to the impact of control and treatment strategies, as well as the monitoring of these drugs. The workers carried multiresistant *S. aureus* and NAS, particularly *S. epidermidis*, a normal inhabitant of human skin and mucous membranes. The location of this bacterium in the milk and skin of animals, as well as in the environment, suggests transmission from humans to animals and the environment. Training in safe and hygienic food handling for staff and the implementation of combined strategies for the control of antimicrobial-resistant bacteria with a One Health approach are recommended. 

## Figures and Tables

**Figure 1 antibiotics-12-01574-f001:**
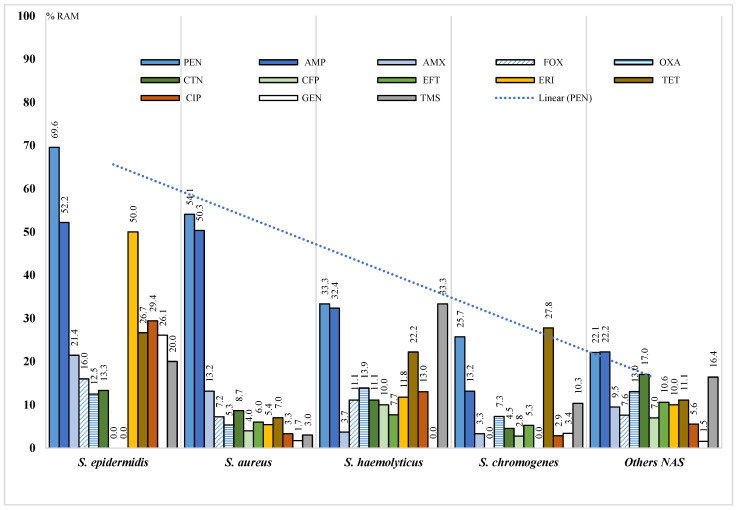
Percentages of antimicrobial resistance (AMR) of the different species of *Staphylococcus* isolated from samples from farms in the Bogotá savanna. FOX: cefoxitin, OXA: oxacillin, PEN: penicillin, AMP: ampicillin, AMX: amoxycillin, CFP: cefoperazone, CTN: cephalothin, EFT: ceftiofur, ERI: erythromycin, TET: tetracycline, CIP: ciprofloxacin, GEN: gentamicin, TMS: trimethoprim/sulfamethoxazole.

**Table 1 antibiotics-12-01574-t001:** Characteristics and pharmacological management of bovine mastitis on dairy farms.

Tier (*n* Cows)	Farm	*S. aureus* *n*	NAS *n*	Total Isolates *n*	Treatment Bovine Mastitis	Treatment Dry Cow	Prescription of Antibiotics	Withdrawal Periods of Antibiotics	MPCP *	Lab **
Small: 10–35	P1	5	5	10	EFT	SP, N	NC	NC	NC	NC
	P2	8	20	28	TY, OT, TET	Not used	NC	C	NC	NC
	P3	0	8	8	Not used	Not used	C	C	C	NC
	P4	1	27	28	OB, AMP	OB, AMP	NC	NC	NC	C
Medium: 36–100	M1	2	6	8	CEF	LN, N	C	C	C	NC
	M2	14	27	41	EFT, SP, N	SP, N	NC	C	NC	NC
	M3	0	5	5	PEN	OB, AMP	C	C	C	NC
	M4	6	11	17	PEN, CT, OT	OB, AMP	C	C	C	NC
	M5	6	25	31	CEF	SP, N, OB, AMP	C	C	NC	NC
Large: >100	G1	10	14	24	AMX	OB, AMP	NC	C	NC	NC
	G2	24	37	61	AMX, TY, OT	OB, AMP	C	C	C	NC
	G3	8	29	37	OB, AMP, CEF	OB, AMP	NC	C	C	C
	G4	33	18	51	OB, AMP, CFL, OT	OB, AMP	NC	C	NC	C

AMX: amoxycillin; CFL: cephalexin; SP: spiramycin; N: neomycin; EFT: ceftiofur; TY: tylosin; OT: oxytetracycline; TET: tetracycline; OB: cloxacillin; AMP: ampicillin; CEF: cefquinone; LN: lincomycin; PEN: penicillin; CT: colistin sulfate. NC: non-compliance; C: compliance. * Mastitis prevention and control program, ** Lab: cultures for bacterial identification and antibiogram tests.

**Table 2 antibiotics-12-01574-t002:** Antibiotic resistance (relative frequencies %) of *Staphylococcus* from samples of 13 farms in the Bogotá savanna.

Species (*n*)	*S. aureus* (152)			NAS (*n* = 197)			Total (349)		
Antibiotics (%)	R	I	S	R	I	S	R	I	S
Cefoxitin	7.2	0	92.8	7.7	0	92.3	7.5	0	92.5
Oxacillin	5.3	0	94.7	11.4	0.5	88.1	8.7	0.3	91.0
Penicillin	54.1	0	45.9	32.4	0.0	67.6	42.0	0	58.0
Ampicillin	50.3	0	49.7	27.0	0.0	73.0	37.2	0	62.8
Amoxycillin	13.2	0	86.8	8.9	0.0	91.1	10.7	0	89.3
Cefoperazone	2.0	2.0	95.9	2.3	3.4	94.3	2.2	2.8	95.1
Cephalothin	8.7	0.0	91.3	10.5	2.9	86.7	9.5	1.3	89.2
Ceftiofur	3.4	2.6	94.0	6.6	1.6	91.8	5.0	2.1	92.9
Erythromycin	3.4	2.0	94.6	9.0	4.3	86.7	6.5	3.3	90.2
Tetracycline	3.5	3.5	92.9	8.8	5.4	85.8	6.5	4.6	88.9
Ciprofloxacin	2.5	0.8	96.7	4.6	4.6	90.8	3.6	2.8	93.6
Gentamicin	1.7	0.0	98.3	4.2	2.8	93.0	3.1	1.5	95.4
Trimethoprim/sulfamethoxazole	2.0	1.0	97.1	16.1	3.5	80.4	10.2	2.4	87.3
Resistant to at least one antibiotic (%)	96/152 (63.2%)			110/197 (55.8%)			206/349 (59.0%)		
Multidrug resistance rate (%)	7/152 (4.6%)			23/197 (11.7%)			30/349 (8.6%)		

**Table 3 antibiotics-12-01574-t003:** Percentage of antimicrobial resistance (AMR) of *Staphylococcus* isolated from different sources on dairy farms in the Bogotá savanna.

Species	Sources	*RI *n* (%)	FOX	OXA	PEN	AMP	AMX	CFP	CTN	EFT	ERI	TET	CIP	GEN	TMS
*S. aureus*	Workers	9/10 (90.0)	40.0	10.0	77.8	66.7	50.0	20.0	0.0	0.0	20.0	66.3	0.0	16.7	10.0
	Environmental	7/7 (100)	0.0	14.3	66.7	100.0	40.0	0.0	33.3	0.0	28.6	0.0	0.0	0.0	--
	QMS	69/117 (59.0)	5.1	4.3	53.4	46.8	9.8	1.8	13.4	5.6	3.6	2.5	3.2	1.1	1.4
	Teats	11/18 (61.1)	5.6	5.6	41.2	41.2	0.0	10.5	0.0	14.3	0.0	--	0.0	0.0	5.9
NAS	Workers	10/15 (61.1)	14.3	7.1	27.3	30.8	42.9	0.0	16.7	0.0	28.6	33.3	0.0	0.0	36.4
	Environmental	19/25 (76.0)	15.4	23.1	34.8	36.0	23.1	4.2	22.2	16.7	23.1	25.0	12.5	13.0	21.1
	QMS	31/59 (52.5)	7.0	8.8	36.2	25.0	6.1	6.3	11.1	8,.1	11.3	6.9	11.4	7.9	14.3
	Teats	50/98 (51.0)	5.1	11.5	30.3	25.6	2.1	6.5	11.1	7.6	9.5	8.5	8.8	4.3	19.7
*Staphylococcus* spp.	Workers	19/25 (76.0)	24.0	8.0	52.4	50.0	41.7	8.7	16.7	0.0	28.0	26.3	0.0	5.6	23.8
	Environmental	26/32 (81.3)	12.9	21.9	39.3	46.9	29.4	3.3	21.7	12.5	21.9	20.0	9.5	14.3	17.4
	QMS	100/176 (56.8)	5.7	5.8	48.0	39.5	8.5	3.1	9.4	6.3	6.0	5.5	6.4	3.1	6.2
	Teats	61/116 (52.6)	5.2	10.5	32.1	28.0	1.9	7.3	8.6	8.8	8.0	13.5	7.2	3.5	17.0

FOX: cefoxitin; OXA: oxacillin; PEN: penicillin; AMP: ampicillin; AMX: amoxycillin; CFP: cefoperazone; CTN: cephalothin; EFT: ceftiofur; ERI: erythromycin; TET: tetracycline; CIP: ciprofloxacin; GEN: gentamicin; TMS: trimethoprim/sulfamethoxazole. *RI: number and percentage of isolates resistant to at least one antibiotic.

**Table 4 antibiotics-12-01574-t004:** Antimicrobial resistance (AMR) profiles of multiresistant *Staphylococcus* isolates from QMSs, teats, workers, and the environment of farms in the Bogotá savanna.

Sample Source	Farm	Species	Resistance Profile
QMS	P2	*S. haemolyticus*	FOX, OXA, PEN, AMP, CFP, CTN, EFT, ERI, TET, TMS
	M1	*S. hyicus*	FOX, OXA, PEN, AMP, CTN, EFT, ERI, TMS
	G2	*S. aureus*	FOX, OXA, PEN, AMP, EFT, ERI
	P4	*S. aureus*	FOX, OXA, CTN, CIP
	P1	*S. aureus*	FOX, PEN. AMP, CIP
	G1	*S. aureus*	OXA, PEN, AMP, AMX, CTN, ERI
	G4	*S. chromogenes*	PEN, AMP, AMX, CFP, GEN
	G1	*S. aureus*	CTN, TET, GEN
	P4	*S. epidermidis*	PEN, ERI, CIP, GEN
	M4	*S. haemolyticus*	PEN, CIP, TMS
	P4	*S. epidermidis*	ERI, CIP, GEN
Teats	P2	*S. haemolyticus*	OXA, PEN, AMP, CFP, CTN, EFT, ERI, TET, CIP, TMS
	P2	*S. equorum*	OXA, PEN, AMP, CFP, CTN, EFT, ERI, TET, CIP, TMS
	G4	*S. xylosus*	FOX, OXA, PEN, AMP, CFP, EFT, CIP
	P2	*S. aureus*	FOX, OXA, PEN, AMP, CFP, TET
	P4	*S. epidermidis*	FOX, PEN, AMP, ERI, CIP, GEN
	P4	*S. epidermidis*	PEN, AMP, ERI, TET, CIP, GEN
	G4	*S. haemolyticus*	OXA, AMP, CFP, TET
	P2	*S. lentus*	OXA, PEN, AMP, CTN, TET
	M5	*S. arlettae*	CFP, EFT, ERI, TET
Workers	G3	*S. aureus*	FOX, OXA, PEN, AMP, AMX, ERI
	P4	*S. epidermidis*	FOX, OXA, PEN, AMP, AMX, TET
	P4	*S. epidermidis*	PEN, AMP, AMX, ERI, TET
	G2	*S. epidermidis*	PEN, AMP, CTN, ERI
	G3	*S. epidermidis*	AMP, ERI, TMS
	M5	*S. lentus*	PEN, ERI, TMS
Environment	P4	*S. warneri*	FOX, OXA, PEN, AMP, CTN, TET, CIP
	G1	*S. epidermidis*	FOX, OXA, PEN, AMP, AMX, CTN, GEN
	P4	*S. epidermidis*	FOX, PEN, AMP, ERI, TET, CIP, GEN
	P4	*S. epidermidis*	OXA, PEN, ERI, GEN, TMS

FOX: cefoxitin; OXA: oxacillin; PEN: penicillin; AMP: ampicillin; AMX: amoxycillin; CFP: cefoperazone; CTN: cephalothin; EFT: ceftiofur; ERI: erythromycin; TET: tetracycline; CIP: ciprofloxacin; GEN: gentamicin; TMS: trimethoprim/sulfamethoxazole.

**Table 5 antibiotics-12-01574-t005:** Genotypes and phenotypes of antimicrobial resistance (AMR) of *S. aureus* obtained from different sources on dairy farms in the Bogotá savanna.

AMR Genotypes and Phenotypes	*n*/Total	Percentage AMR (%)	AMR Genotype/Phenotype *K* ** (CI95%) (Observed Agreement %)
*blaZ* gene	120/143	83.9	0.03 (−0.09, 0.16) (53.6)
K.B. penicillins *	79/152	52.5	
*mecA* gene	17/140	12.1	0.16 (0.0, 0.3) (85.7)
K.B. FOX	11/152	7.2	
*ermC* gene	34/145	23.5	0.07 (−0.04, 0.19) (75.4)
K.B. ERI	8/148	5.4	
*tetK* gene	18/140	12.9	0.14 (−0.04, 0.32) (86.2)
K.B. TET	8/113	7.1	
*aacA-aphD* gene	71/146	48.6	−0.06 (−0.15, 0.03) (54.2%)
K.B. GEN	2/118	1.7	

* Penicillins: penicillin, oxacillin, ampicillin, amoxycillin. KB: Kirby–Bauer; FOX: cefoxitin; TET: tetracycline; ERI: erythromycin; GEN: gentamicin. ** *K*: Cohen’s Kappa coefficient.

**Table 6 antibiotics-12-01574-t006:** List of primers used in the study for detection of AMR genes in *Staphylococcus aureus* isolates from dairy farms.

PCR/ Program	Genes	Primer Sequence	Amplicon Size (bp)	Reference
Simple ^1^	*mecA*	5′-GTAGAAATGACTGAACGTCCGATAA-3′ 5′-CCA ATT CCA CAT TGT TTC GGT CTAA-3′	310	[65]
Multiplex ^2^	*blaZ*	5′-ACTTCAACACCTGCTGCTTTC-3′ 5′-TGACCACTTTTATCAGCAACC-3′	173	[66]
	*tetK*	5′-GTAGCGACAATAGGTAATAGT-3′ 5′-GTAGTGACAATAAACCTCCTA-3′	360	
	*ermB*	5′-CTATCTGATTGTTGAAGAAGGATT-3′ 5′-GTTTACTCTTGGTTTAGGATGAAA-3′	142	
Simple ^3^	*aacA-aphD*	5′-GAAGTACGCAGAAGAGA-3′ 5′-ACATGGCAAGCTCTAGGA-3′	491	[66]

^1^ 94 °C for 4 min; then 30 cycles of 94 °C for 45 s, 50 °C for 45 s, 72 °C for 1 min; and 72 °C for 2 min; ^2^ 95 °C for 3 min; then 30 cycles of 95 °C for 30 sec, 54 °C for 30 s, 72 °C for 30 s; and 72 °C for 4 min; ^3^ 95 °C for 5 min; then 30 cycles of 95 °C for 2 min, 54 °C for 1 min, 72 °C for 1 min; and 72 °C for 7 min.

## Data Availability

Not applicable.

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
