# Peer review of "Antimicrobial Resistance Profiles of Staphylococcus Isolated from Cows with Subclinical Mastitis: Do Strains from the Environment and from Humans Contribute to the Dissemination of Resistance among Bacteria on Dairy Farms in Colombia?"

_antibiotics, 2023, doi:10.3390/antibiotics12111574_

Round 1

Reviewer 1 Report

Comments and Suggestions for Authors

Dear authors,

The present paper is related with one of the main dairy problems in all the world and the included information is interesting and clear to get an idea the how is AMR in Colombian dairy farms.

I suggest that you could describe with more details the study region (total bovine population) to known the impact of this study.

You could find some comments and observation in the attached file.

Author Response

Thank you very much for taking the time to review this manuscript. Please find the detailed responses below and the corresponding revisions in track changes in the re-submitted files

1. I suggest that you could describe with more detail the study region (total bovine population) to know the impact of this study 

Answer: The statement regarding the study description has been modified.

  1. You could find some comments and observations in the attached file

Answer: Thank you for your comments, minor changes have been made to the document.

Reviewer 2 Report

Comments and Suggestions for Authors

In my opinion, the manuscript submitted for review is very interesting and I like the idea of it. I have no major objections to the work, because its content corresponds to its title, and the scope of research is within the scope of the journal. Therefore, I believe that the work can be accepted for publication. However, please check the bibliography - double numbering.

Author Response

Thank you very much for taking the time to review this manuscript

1. In my opinion, the manuscript submitted for review is very interesting and I like the idea of it. I have no major objections to the work, because its content corresponds to its title, and the scope of research is within the scope of the journal. Therefore, I believe that the work can be accepted for publication. However, please check the bibliography - double numbering.

Answer: Thank you for your fruitful comments, the bibliography has been checked.

Reviewer 3 Report

Comments and Suggestions for Authors

The paper presented by Martha Fabiola Rodríguez and her colleagues provides a comprehensive description of the antimicrobial resistance profiles of Staphylococcus isolated from cows with subclinical mastitis on Colombian farms. The overall structure of the paper was well-suited and easy to understand. I don't have any further comments to add.

Author Response

Thank you very much for taking the time to review this manuscript.

1. The paper presented by Martha Fabiola Rodríguez and her colleagues provides a comprehensive description of the antimicrobial resistance profiles of Staphylococcus isolated from cows with subclinical mastitis on Colombian farms. The overall structure of the paper was well-suited and easy to understand. I don't have any further comments to add.

Answer: Thank you for your comments

Reviewer 4 Report

Comments and Suggestions for Authors

In this study by Rodríguez, Gomez and Ceballos-Garzon (2023), the authors investigated the phenotypic and genotypic antimicrobial resistance profile of different Staphylococcus species isolated from cows with subclinical mastitis. The work is relevant and pertinent to the scope of the journal. However, some minor points need to be improved before this manuscript can be published. Please see below:

1) line 15: dissemination of this bacteria (?)

2) line 24: Please enter the concentration of the antimicrobials used.

3) line 87, 96, 136, 144, 150, 180, 187, 193 and wherever else necessary throughout the manuscript.: please use italics.

4) Table 2: Please include the meaning of bold and/or underlined values in the figure captions, tables and figures should allow them to be interpreted independently of the rest of the text.

5) line 310: how were the microorganisms isolated?

6) line 311: the quotation mentioned seems to be missing.

7) line 330: please correct the spelling of the numbers.

8) lines 382-384: this period is not relevant to the conclusion, please stick to the findings of this work, I suggest removing it.

Author Response

Thank you very much for taking the time to review this manuscript. Please find the detailed responses below and the corresponding revisions corrections in track changes in the re-submitted files

In this study by Rodríguez, Gomez and Ceballos-Garzon (2023), the authors investigated the phenotypic and genotypic antimicrobial resistance profile of different Staphylococcus species isolated from cows with subclinical mastitis. The work is relevant and pertinent to the scope of the journal. However, some minor points need to be improved before this manuscript can be published. Please see below:

  1. line 15: dissemination of this bacteria (?)

Answer: The sentence has been modified (line 15).

  1. line 24: Please enter the concentration of the antimicrobials used.

Answer: The concentration of the antimicrobials used has been included (line 24).

  1. line 87, 96, 136, 144, 150, 180, 187, 193 and wherever else necessary throughout the manuscript.: please use italics.

Answer: The changes have been made in the document (line 88, 97, 138, 146, 152, 181, 187, 193)

  1. Table 2: Please include the meaning of bold and/or underlined values in the figure captions, tables and figures should allow them to be interpreted independently of the rest of the text.

Answer: The changes have been made in table 2. No values were left bold and/or underlined

5) line 310: how were the microorganisms isolated?

Answer: The statement regarding microorganisms’ isolation has been included (lines 317-318).

6) line 311: the quotation mentioned seems to be missing.

Answer: The Reference has been added (lines 319,326)

7) line 330: please correct the spelling of the numbers.

Answer: The changes have been made in the document (line 338)

8) lines 382-384: this period is not relevant to the conclusion, please stick to the findings of this work, I suggest removing it.

Answer: The changes have been made in the document (line 391)